# Adverse Maternal Environments Perturb Hepatic DNA Methylome and Transcriptome Prior to the Adult-Onset Non-Alcoholic Fatty Liver Disease in Mouse Offspring

**DOI:** 10.3390/nu15092167

**Published:** 2023-04-30

**Authors:** Qi Fu, Warren A. Cheung, Amber V. Majnik, Xingrao Ke, Tomi Pastinen, Robert H. Lane

**Affiliations:** 1Department of Research Administration, Children’s Mercy Hospital, Kansas City, MO 64108, USA; 2Genomic Medicine Center, Children’s Mercy Hospital, Kansas City, MO 64108, USA; 3Department of Pediatrics, Medical College of Wisconsin, 8701 W Watertown Plank Rd., Milwaukee, WI 53226, USA; 4Department of Administration, Children’s Mercy Hospital, Kansas City, MO 64108, USA

**Keywords:** NAFLD, methylation, DNA, epigenetics, liver, development, maternal, early life stress, diet, perinatal

## Abstract

Exposure to adverse early-life environments (AME) increases the incidence of developing adult-onset non-alcoholic fatty liver disease (NAFLD). DNA methylation has been postulated to link AME and late-onset diseases. This study aimed to investigate whether and to what extent the hepatic DNA methylome was perturbed prior to the development of NAFLD in offspring exposed to AME in mice. AME constituted maternal Western diet and late-gestational stress. Male offspring livers at birth (d0) and weaning (d21) were used for evaluating the DNA methylome and transcriptome using the reduced representation of bisulfite sequencing and RNA-seq, respectively. We found AME caused 5879 differentially methylated regions (DMRs) and zero differentially expressed genes (DEGs) at d0 and 2970 and 123, respectively, at d21. The majority of the DMRs were distal to gene transcription start sites and did not correlate with DEGs. The DEGs at d21 were significantly enriched in GO biological processes characteristic of liver metabolic functions. In conclusion, AME drove changes in the hepatic DNA methylome, which preceded perturbations in the hepatic metabolic transcriptome, which preceded the onset of NAFLD. We speculate that subtle impacts on dynamic enhancers lead to long-range regulatory changes that manifest over time as gene network alternations and increase the incidence of NAFLD later in life.

## 1. Introduction

Non-alcoholic fatty liver disease (NAFLD) is the most common liver disease in the United States and worldwide. Its prevalence is steadily increasing and occurs in ~30% of Americans [1]. The incidence of adult NAFLD increases following exposure to early-life adverse experiences, such as maternal Western-style diet and maternal life stress [2,3]. In animal studies, a priming effect of adverse early-life environments on the development of NAFLD later in life has been reported by multiple groups [2,4,5]. However, the mechanisms linking early-life adverse environments to adult-onset NAFLD remain elusive.

Epigenetics, such as DNA methylation, has been shown to link early life environment and long-term illness and has emerged as a prime candidate for mediating such long-term ‘programming’ effects [6,7,8,9,10]. DNA methylation is an epigenetic modification that is largely restricted to CpG dinucleotides and serves critical functions, such as transcriptional regulation. DNA methylation in promoter regions and gene bodies is negatively and positively correlated with gene transcription levels, respectively [11,12]. Similarly, active distal regulatory elements show hypomethylation, while most non-coding DNA is hypermethylated in somatic cells [13]. Importantly, DNA methylation is actively regulated during embryonic and early postnatal developmental periods [14,15], during which the major organ systems emerge and develop, thereby becoming vulnerable to environmental perturbations [16].

The perinatal period, especially the in utero period, is the developmental window when epigenetic status has the greatest plasticity and is hence most vulnerable to environmental insults [17]. Animal model studies have demonstrated perturbed hepatic DNA methylation during this period, mainly at gene-specific levels [18,19]. However, it remains unclear if an adverse early-life environment causes genome-wide DNA methylome changes in the perinatal liver. In terms of adult-onset NAFLD, DNA methylation/methylome perturbation has been reported at the time of adult-onset NAFLD caused by the adverse early-life environment [2,5,20,21]. However, the timing of epigenetic changes may be critical in mediating long-term liver injury, and the timing or sequence of events is currently unknown. We subsequently hypothesized that an adverse early-life environment would perturb the hepatic DNA methylome prior to adult-onset NAFLD. We further hypothesized that DNA methylome changes would coincide with perturbations of the hepatic transcriptome.

To test these hypotheses, we used a previously established mouse model of adult-onset NAFLD that evolves from an adverse early-life environment [2]. In this model, the adverse early-life environment (AME) featured a maternal Western diet from five weeks prior to pregnancy through lactation and maternal chronic moderate life stress during the last week of pregnancy. In this model, a significantly higher number of male offspring exposed to AME developed mild but significant hepatic steatosis in adulthood, even though they were on a low-fat control diet for their entire postweaning lives. The DNA methylome was studied using reduced representation bisulfite sequencing (RRBS) at single-base resolution, and the transcriptome was studied using RNA-Seq at birth (the endpoint of the in utero environment) and at weaning (the early postnatal period).

## 2. Materials and Methods

### 2.1. Animals

All animal procedures conformed to the National Institutes of Health Guide for the Care and Use of Laboratory Animals and were approved by the Medical College of Wisconsin Institutional Animal Care and Use Committee (Protocol No. 00003557). This mouse model of AME has been previously published by our group [2,22]. Briefly, female C57BL/6J mice were maintained with standard light cycles at a constant temperature of 23 °C, with food and water available ad libitum. The mice were randomly allocated to either an experimental Western-style diet (WD) or a control diet (CD) (n = 5/group) [2]. The WD contained increased fat (40% compared with 10% in kcal), sucrose (29.1% compared with 0% in kcal), and cholesterol [0.15% compared with 0% (*wt/wt*)] compared with the CD (D14020502, Research Diets). The diets were provided for 5 weeks before the females were mated with males [males received Laboratory Rodent Diet (5L0D, LabDiet)] and were continued throughout pregnancy and lactation.

Dams fed by the WD also experienced a “stressed” environment during the last third of pregnancy. The combination of chronic WD and gestational stress is designated as AME [2,22]. Briefly, the stressed environment consisted of daily random environmental changes as well as a static change in the maternal environment consisting of one-third of the standard amount of bedding from embryonic day (E)13 to E19. The daily random environmental changes included altered light cycles on three nonconsecutive days, three repeat cage changes throughout the day on E15, and the short-term introduction of a novel object in the cage for a day. Control (Con) dams consumed CD and were not exposed to stress [2,22]. Day 0 (d0) offspring were harvested by C-section from pregnant females (n = 5/group) after 21 days of gestation. Body weights were recorded, and livers were quickly removed, weighed, flash frozen in liquid nitrogen, and stored at −80 °C. Tails were collected for sex determination. Another set of pregnant female mice (n = 5/group) from both the Con and AME groups were allowed to deliver naturally. On postnatal day 5, litters were randomly culled to six. The offspring stayed with their dam until d21. At d21 (n = 5 litters of each group), mice were anesthetized with isoflurane and killed by decapitation after 5 h of fasting. Body weights were recorded. Livers were harvested as above-mentioned. In this model, the male offspring developed significantly more hepatic steatosis than the control male offspring at 17 weeks of life, despite the consumption of CD for 14 weeks [2]. We used the livers of the male mice from both d0 and d21 for this current study.

#### 2.1.1. Sex Determination

PCR-based sex determination was based on the method published by McClive, P.J. et al. [23]. The d0 mouse tail was digested in 50 uL of complete digestion buffer [50 mM KCl, 10 mM Tris-HCl (pH 8.3), 0.1 mg/mL gelatin, 0.45% NP40, 0.45% Tween-20, 0.4 mg/mL proteinase K] for 15 min at 55 °C, followed by 10 min at 95 °C. A total of 1 ul was used for PCR using the forward primer 5′GATGATTTGAGTGGAAATGTGAGGTA and the reverse primer 5′CTTATGTTTATAGGCATGCACCATGTA. The PCR condition was 95 °C for 10 min, followed by 35 cycles of 95 °C for 15 s, 57 °C for 30 s, and 72 °C for 30 s. One 280 bp band on an agarose gel electrophoresis indicates a male sex; 1–3 bands around 685 bp indicate a female.

#### 2.1.2. Hepatic TG Quantification

The liver tissues from d21 male mice were ground in liquid nitrogen. One portion of the ground liver was weighed and used for TG isolation and quantification. TG isolation and quantification were performed using the Triglyceride Quantification Kit (MAK266, Sigma-Aldrich, St. Louis, MO, USA) following the manufacturer’s manual. The hepatic TG levels were expressed as mg/gram of liver tissue. 

### 2.2. Reduced Representation Bisulfite Sequencing (RRBS)

Ground livers were used for genomic DNA isolation using the DNeasy Blood & Tissue Kit (Qiagen, Hilden, Germany), including RNase treatment. DNA quantity and purity were estimated spectrophotometrically. RRBS (n = 5 for each age) was performed at the Genomic Sciences and Precision Medicine Center (GSPMC), Medical College of Wisconsin. RRBS libraries were prepared using 250 ng of genomic DNA, digested with Msp1 (New England Biolabs (NEB), Ipswich, MA, USA, R0106M), and purified using the QIAquick Nucleotide Removal Kit (Qiagen, 28004). End-repair A tailing was performed (NEB, M0212L), and TruSeq methylated-indexed adaptors (Illumina, San Diego, CA, USA, 15025064) were ligated with T4 DNA ligase (NEB, M0202L). Size selection was performed with AMPure XP beads (Beckman Coulter, Brea, CA, USA, A63882). Bisulfite conversion was performed using the EpiTect Plus Bisulfite Kit (Qiagen) as recommended by the manufacturer. Following bisulfite treatment, the DNA was purified as directed and amplified using Pfu Turbo Cx HotStart DNA Polymerase (Agilent Technologies, Santa Clara, CA, USA, 600414). Library quantification was performed using the Qubit dsDNA HS Assay Kit (Life Technologies, Carlsbad, CA, USA, Q32854) and the Bioanalyzer DNA 1000 Kit (Agilent Technologies, 5067-1504). The final libraries from RRBS were prepared for sequencing as per the manufacturer’s instructions in the Illumina cBot and HiSeq Paired-End Cluster Kit v3. Samples were sequenced at 51 bp paired-end reads using Illumina HiSeq 2000 with TruSeq SBS sequencing kit v3. Base calling was performed using Illumina’s RTA version 2.7.3.

### 2.3. RNA-Seq

Ground livers from d0 and d21 male (n = 5) mice were used for total RNA isolation by using a miRNeasy Mini Kit (Qiagen) following the manufacturer’s instructions, including an on-column DNase I treatment. RNA was quantified spectrophotometrically. The integrity of RNA was assessed with an Agilent 2100 bioanalyzer in conjunction with the RNA 6000 Nano Kit (Agilent), and a RNA integrity number greater than or equal to nine was used. RNA-Seq was performed at the GSPMC, Medical College of Wisconsin. RNA libraries were prepared by using 200 ng of total RNA according to the manufacturer’s instructions for the Illumina TruSeq RNA v2 kit (Illumina, San Diego, CA, USA), and sequencing was completed on the Illumina HiSeq-2000 with 101 bp paired-end reads. Base calling is performed using Illumina’s RTA version 2.5.2.

### 2.4. Real-Time Reverse Transcriptase (RT) -PCR

DEGs were validated using real-time RT-PCR. cDNA was synthesized using a high-capacity cDNA reverse transcription kit (4368814, Thermo Fisher Scientific, Waltham, MA, USA). Real-time RT-PCR was performed as described earlier [2]. Target primers and probes are shown in Appendix A. Hypoxanthine guanine phosphoribosyl transferase (Hprt) was used as the housekeeping gene [2]. PCR conditions and calculations of mRNA expression were performed as demonstrated previously [2].

### 2.5. Bioinformatics Analysis and Data Integration of RRBS with RNA-Seq

Differentially methylated cytosines (DMCs) were identified by analyzing genome annotations from ENCODE/GENCODE Version M11 (Ensembl 86) [24]. Then, the DMCs (*p* < 0.001; Δme 10%) were aligned against the database of mouse Encyclopedia of DNA Elements (mENCODE) to find the ones that overlapped with the cis-regulatory elements (cCREs) [25]. Briefly, the cCREs were categorized into (1) promoter-like elements (PLS), which were further divided into canonical promoter-like elements, which fall within 200bp of an annotated GENCODE transcription start site (TSS), and other histones (H)3 lysine (K)4 trimethylation (me)3 elements, which had features similar to the PLS but were not within 200bp of the TSS. (2) Enhancer-like elements (ELS), which were subdivided into proximal ELS (pELS) and distal ELS (dELS) defined by DNase hypersensitivity and high H3K27acetylation (ac) signals with their distances to TSS within or more than 2kb, respectively. (3) CCCTC-binding factor (CTCF)-only elements [25].

### 2.6. GO Term Analysis

We used the Genomic Regions Enrichment of Annotations Tool (GREAT) version 4.0.4 [26,27] for all DMR-related GO term analyses. DMRs were analyzed against mouse genome mm10 using 121,978 CpGs randomly generated from ~6 million CpG (2% of all measured) sites from the mouse genome as a background dataset. Region-gene association data were binned by orientation and distance to the single nearest gene TSS within 1 Mb into 0–5 kb, 5–50 kb, 50–500 kb, and >500 kb at both upstream and downstream of the TSSs.

### 2.7. Statistical Analysis

Statistical analysis was performed using GraphPad Prism 8 software (GraphPad Software). The validation of the mRNA expression was analyzed using a Mann–Whitney test. The level of significance was set at *p* < 0.05 for all statistical tests.

## 3. Results

### 3.1. AME Male Offspring Did Not Develop Hepatic Steatosis but Were Significantly Heavier Than the Age-Matched Controls at d21

AME did not impact body weight (BW) at d0. However, by d21, AME significantly increased the BW of the male offspring compared to that of the Con group (Appendix A).

Paralleling the BW changes, liver weights were significantly higher in d21 AME relative to d21 Con (0.456 ± 0.051 g in AME vs. 0.371 ± 0.048 g in Con, *p* < 0.05). However, the ratio of liver weight/BW (0.0431 ± 0.003 in AME vs. 0.0430 ± 0.003 in Con) remained identical between the two groups. 

At the age of d21, AME slightly, but not significantly, increased hepatic TG content compared to the Con in males (Appendix A). No significant histological changes were found in male livers at d21.

### 3.2. AME Perturbed the DNA Methylome at Both d0 and d21

RRBS was performed for liver methylomes in the four groups. Non-supervised principal component analysis (PCA) showed a clear separation between the two ages of d0 and d21 (Appendix A). RRBS covered 1,750,178–2,302,225 CpG sites per sample. Among them, 49,143 differentially methylated regions (DMRs) were found between Con-d21 and Con-d0 and 56,796 between AME-d21 and AME-d0 (Figure 1A,B). To determine if these DMRs were potentially functional, the DMRs were analyzed against the mouse Encyclopedia of DNA Elements (mENCODE) Consortium [25,28]. A total of 23.4% (11,478 out of 49,143) and 21.0% (11,940 out of 56,796) overlapped with putative cis-regulatory elements (cCREs) in the Con and AME, respectively (Figure 1A,B). These results indicated that up to 21% of the DMRs were potentially functional cCREs.

The developmental changes in the DNA methylome from d0 to d21 are complex due to developmental changes in the landscape of the methylome in individual liver cells that undergo differentiation and shifts in the relative abundance of different cell types that compose a liver. Therefore, we selected to investigate the effect of AME on DMRs between groups of the same age, specifically by comparing changes between AME-d0 and Con-d0 and between AME-d21 and Con-d21.

At d0, AME resulted in 5879 DMRs, including 16% of those being proximal to genes and 84% in distal regions. When analyzed against mENCODE, 996 DMRs overlapped with the putative cCREs [28] (Figure 2A). At d21, 2970 DMRs were identified (16% proximal and 84% distal), and 880 overlapped with mENCODE cCREs (Figure 2B). The calculations of Δme (the difference in average methylation between groups) for each DMR showed a significantly larger proportion of hypomethylated DMRs among the DMRs observed at d0 (71% hypomethylated, binomial *p*-value < 2 × 10^−6^) (Figure 2C,D). By d21, there was more of a balance between hypo- and hyper-methylated DMRs (48% hypomethylated, binomial *p*-value = 0.02). The average magnitude of the change in methylation was similar at 18% at d0 and 17% at d21 (Figure 2E,F).

These thousands of AME-induced DMRs at d0 and d21 are extremely exciting because the livers were not steatotic at either age. This indicated that AME-induced changes in the DNA methylome during critical developmental windows preceded hepatic steatosis.

Noticeably, the majority of the cCRE-overlapped DMRs were in the class of distal enhancer-like elements (dELSs) (72.3% at d0 and 70.6% at d21, respectively) (Figure 2A,B). To reveal the potential functional significance of these distal DMRs, we used the Genomic Regions Enrichment of Annotations Tool (GREAT) [26,27] to analyze the Gene Ontology (GO) terms of these DMRs.

We compared the genome-wide distribution of the DMRs against a random background genomic dataset using the GREAT tool. The DMRs, which were binned based on distances from the nearest TSSs, showed enrichment for all binned distances up to 50 kb to 500 kb away from the nearest TSSs, whereas TSS-proximal DMRs (0–5 kb) were significantly depleted as compared to the background of 120,788 randomly chosen CpGs at both d0 and d21. Specifically, we observed that 11% of DMRs in AME-d0 vs. Con-d0 (*p*-value < 1 × 10^−6^, χ^2^(1df) = 669) (Figure 3A), and 16% in AME-d21 vs. Con-d21 (*p*-value < 1 × 10^−6^, χ^2^(1df) = 156) (Figure 3B) were located at TSS (0–5 kb bin) compared to the 26% found in the background CpG distribution.

Due to the large number of DMRs, we then decided to focus on the AME effects on DNA methylome changes by focusing on the 312 DMRs shared by the AME-d0 vs. Con-d0 and the AME-d21 vs. Con-d21 (Figure 3C,D). These shared DMRs were showing a strong depletion of those proximal to TSSs compared to the background dataset (10.6% vs. 26%, *p*-value < 1 × 10^−6^, χ^2^(1df) = 39) (Figure 3C,D).

As the majority of the DMRs were significantly distal from proximal promoters, we used the GREAT tool to explore if these DMRs were significantly enriched in any biological processes or molecular functions. A total of 63 genes were within 10kb upstream or downstream of the single nearest TSS. No Gene Ontology terms were significantly enriched in the DMRs compared to the background dataset.

Considering that RRBS only covers around 10% of CpGs in the genome, with stronger representation for proximal regulatory (promoter) regions, the number of AME-resulted DMRs could be substantially greater than the numbers currently revealed in this study. Therefore, we would predict many DEGs should be found at both ages, based on the canonical relationship between DNA methylation at the regulatory regions and gene transcription (negative correlation between CpG methylation and gene transcription). The DEGs between AME-d0 and Con-d0 may indicate the impact of maternal environment on offspring transcriptome during in utero development, while the DEGs between AME-d21 and Con-d21 may indicate the cumulative impact of both the in utero and early postnatal life environments on offspring hepatic transcriptome.

### 3.3. AME Did Not Impact Hepatic Transcriptome at d0 despite the Large Number of DMRs

RNA-seq was performed on the whole liver transcriptome. Non-supervised PCA analysis recapitulated the expected pattern; the major difference was d0 to d21 (Appendix A). 

In transcriptome analyses between AME and controls at d0, no single statistically significant DEG was revealed. Concordantly, none of the DMRs between AME-d0 and Con-d0 were near TSS. These data showed extreme concentrations of the 5879 DMRs between AME-d0 and Con-d0 at non-promoter regulatory DNA. To ensure that this was not a false negative, we aligned the DMRs and DEGs between d0 and d21 in the control group. Specifically, we compared Con-d0 with Con-d21 since there were 5202 DEGs between the two groups. Among the 49,143 DMRs, 8861 were within 25 kb of a DEG, and 5616 had a canonical relationship between CpG methylation and DEG gene expression. This developmental correlation between DMRs and DEGs indicated that the zero DEG and noncorrelation between DEGs and DMRs between AME-d0 and Con-d0 were probably not false negatives. This unperturbed hepatic transcriptome at d0 by AME could indicate that the hepatic transcriptome is hardwired and protected by the in utero environment.

### 3.4. At d21, AME Perturbed Hepatic Transcriptome, Which Did Not Correlate with DMRs

At d21, AME resulted in 123 DEGs, with 36 upregulations and 87 downregulations when compared with AME-d21 and Con-d21 (Figure 4A–C).

Consistent with the d0 data, only 11 out of the 123 DEGs were within 25 kb of DMRs, and only 5 in the canonical relationship between DNA methylation and gene expression (negative correlation). This unrelatedness between DMRs and DEGs could be due to the majority of the DMRs being distal to TSSs.

### 3.5. AME Significantly Perturbed Genes on Metabolic Pathways at d21

The AME-downregulated genes were significantly enriched in multiple GO terms (Figure 4D). We used an adjusted p-value cut-off of 0.0001 to focus on the strongest signals and top 10 GO biological processes. These processes included cholesterol biosynthesis, cholesterol metabolism with Bloch and Kandutsch–Russell pathways, metabolism of lipids, lipid biosynthetic processes, metabolism of steroids, sterol biosynthetic processes, secondary alcohol biosynthetic processes, cholesterol biosynthetic processes, steroid metabolic processes, and alcohol metabolic processes. The genes involved in these GO terms included Lpin1, Fasn, etc.

The AME-upregulated genes were significantly enriched in 11 GO terms when the adjusted p-value cut-off of 0.0001 was used. These GO biological processes included unsaturated fatty acid metabolic processes, steroid hormone biosynthesis, retinol metabolism, arachidonic acid metabolism, chemical carcinogenesis, the epoxygenase P450 pathway, monocarboxylic acid metabolic processes, linoleic acid metabolism, icosanoid biosynthetic processes, carboxylic acid biosynthetic processes, and organic acid biosynthetic processes that were significantly enriched (Figure 4E). The genes involved in these pathways, such as Il1b, Ces1g, and Elovl3, have been reported to be related to NAFLD and have been validated [29,30,31] (Figure 4F). 

Taken together, these data indicated that AME perturbation of the DNA methylome preceded the perturbation of the hepatic metabolic transcriptome, which preceded the onset of hepatic steatosis.

## 4. Discussion

This study investigated the hepatic genome-wide DNA methylome and transcriptome at birth and at weaning in the mice's offspring who experienced an adverse early-life environment and would develop NAFLD at 17 weeks of life. The overarching finding of this study was that the maternal environment drove changes in the hepatic DNA methylome, which preceded perturbations in the hepatic metabolic transcriptome, which preceded the onset of NAFLD. Specifically, we found the following results: (1) AME impacted the DNA methylome prenatally and postnatally before the development of liver steatosis. (2) The majority of the DMRs were distal to TSSs and did not correlate with DEGs. (3) AME-induced DEGs at d21 were significantly enriched in multiple GO biological processes, such as fatty acid metabolic process, steroid hormone synthesis, cholesterol biosynthesis, etc. We speculate that AME leads to changes in the activity of long-range enhancers in offspring with subtle alterations in hepatic gene networks that increase the incidence of NAFLD later in life.

In this study, we used an established AME murine model of NAFLD [2]. In this model, at week 17 of life, the offspring develop mild but significant hepatic steatosis despite having been on a control diet for their entire postweaning lives. A postweaning Western diet exacerbated the priming effect of the maternal environment, resulting in a more severe steatosis phenotype. This priming effect of the maternal environment has also been reported by other groups using maternal diet models [4,5]. These models also have a metabolic profile similar to that observed in human NAFLD, including obesity and insulin resistance [4,5]. Therefore, our findings on the DNA methylome and transcriptome in this mouse model are translationally relevant in studying the pathogenesis of NAFLD, which has an early-life origin.

Murine models have also shown links between hepatic DNA methylome perturbation and the early-life origin of adult-onset NAFLD. Gutierrez Sanchez et al. established a mouse nonalcoholic steatohepatitis (NASH) model with a maternal diet of high fat, high fructose/glucose, and high cholesterol (FFC) during gestation and lactation. Using RRBS, the study found that the maternal FFC diet alone, without the postweaning FFC diet, resulted in a total of 283 DMRs when compared with the offspring of the maternal control diet group at the time of NAFLD (10 weeks of life). Among the DMRs, only 11 (3.9%) were correlated with DEGs (RNA-Seq) [5]. In another study, Wankhade UD et al., using a murine model of NAFLD induced by a maternal high-fat diet, used RRBS and RNA-Seq to reveal that maternal diet alone resulted in 82 DMRs in offspring livers at 17 weeks of age when NAFLD occurred. Among the 82 DMRs, ~78% overlapped a gene [20]. These data indicate that maternal diets had long-lasting effects on the hepatic DNA methylome into adulthood when NAFLD occurred. However, it is still unknown to what extent the hepatic DNA methylome is perturbed by maternal environments before the development of offspring NAFLD, despite previous studies. To the best of our knowledge, our study is among the first to demonstrate that the impact of AME on the hepatic DNA methylome is observable during a critical developmental period, long before the onset of NAFLD in adulthood.

There is a body of evidence that establishes changes in the early-life DNA methylome in other chronic diseases with a late onset, supporting its role in the developmental origin of adult diseases (DOHaD). In human studies, DNA methylation changes in offspring blood and placenta have been associated with adverse maternal environments. The Human Early Life Exposome Project used a multi-center cohort of 1301 mother–child pairs and 91 different pregnancy environmental exposures to demonstrate that maternal levels of molybdenum were correlated with the blood methylation levels of 72 CpGs, which were persistent until childhood [32]. The Pregnancy and Childhood Epigenetics Consortium did a large epigenome-wide association study and associated over 400 placental DMRs with maternal cigarette smoking [33]. However, these DMRs rarely overlapped with those found in the cord blood of the same maternal-fetal dyad [34]. These human association studies support the hypothesis that early-life DNA methylome perturbation precedes the onset of later-life health issues in an organ- and early-life environment-specific manner. Our current study adds to the literature by revealing thousands of early-life environments with vulnerable hepatic DMRs, including the 21% that overlapped with mENCODE cCREs. Some of the differences in the scale of these discoveries may be because animal studies such as ours can control environmental conditions in a way human studies cannot. Taken together, both the literature and our study indicate that adverse early-life environments that perturb the DNA methylome precede the onset of adult disease. Future work will be necessary to understand the functions of these DMRs, especially the distal DMRs.

Distal (non-TSS) DNA CpG methylation imposes regulatory functions on high-order chromatin structure and distal regulatory element function [35]. Most of the dynamic (variable) methylome in humans is distal to TSS in differentiating cell lineages and developmental states [36]. We are only beginning to understand the role of distal CpG methylation in disease [19,37,38,39]. Park YJ et al. have demonstrated that adipocyte DNA methylation synergistically cooperates with cis-regulatory networks and chromosome structures to establish distinct gene expression profiles that confer metabolic features to adipocytes [39]. Ankill et al. have shown that proliferation in breast cancer is linked to the loss of methylation at specific enhancers and transcription factor binding sites, thereby affecting gene activation through chromatin looping [40]. To explore if the AME-associated DMRs found in this study have long-range regulatory potential, we analyzed our AME-resulted DMRs against a publicly available promoter capture Hi-C dataset derived from C57BL/6 mice fed either a lipid-rich diet (HF) or a carbohydrate-rich diet (LF) [41]. Enrichment analyses on the extracted list of genes linking distal elements to the closest TSSs identified the term “fatty liver disease” for the d0 DMRs (Appendix A). Further studies will be required to understand if any of the DMRs found in this study play a role in long-range chromatin regulation.

One of the limitations of this study is the nature of the RRBS technique. RRBS is a high-throughput sequencing strategy that enriches its libraries by digesting genomic DNA with restriction endonucleases that are specific for CpG-containing motifs. It provides enhanced coverage for the CpG dinucleotides and yields single base pair resolution data within regions of interest, including CpG islands, promoters, and enhancer elements [42,43]. It captures around 10% of CpG sites within mammalian genomes, with up to a 30-fold reduction in the number of fragments sequenced in comparison to whole genome bisulfite sequencing (WGBS) [42]. However, the technique is naturally biased towards CpG-rich regulatory elements, capturing a higher proportion of proximal vs. distal regulatory elements. Consequently, our findings are a conservative estimate of differences in distal dynamic regulatory elements. Another limitation was that we did not differentiate the effects of the maternal Western diet and maternal stress on the offspring NAFLD. However, we consider this study relevant because the combination of calorie-dense diet and stress is a environment many peri-pregnancy women live in.

## 5. Conclusions

The link between early-life DNA methylome and DOHaD in response to environmental cues is still at an early stage of deciphering. This study has revealed that changes in epigenetics anticipate metabolic perturbations, which precede the onset of NAFLD in adulthood. We believe that with the exponential increase in genome-wide spatial and temporal mapping/cataloging of epigenomes, studies similar to ours will collectively improve our understanding of epigenome changes in response to environmental cues, thereby contributing to the knowledge of DOHaD. From the standpoint of DOHaD, characterization of the “DOHaD epigenome” could provide baseline work for potential epigenome intervention for the prevention of DOHaD.

## Figures and Tables

**Figure 1 nutrients-15-02167-f001:**
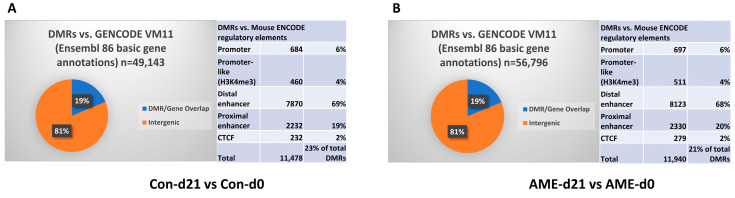
Effect of development on DMRs in (**A**) control livers between Con-d21 and Con-d0 and (**B**) AME livers between AME-d21 and AME-d0. The pie graphs show the percentage of DMRs overlapping genes in the intergenic regions by analyzing the DMRs against mouse gene annotations [GENCODE Version M11 (Ensembl 86)]. The tables show the number and percentage of DMRs overlapping mouse ENCODE cis-regulatory elements, which are categorized into promoter, promoter-like (H3K4me3), distal enhancer, proximal enhancer, and CTCF binding sites. AME, adverse maternal environment; Con, control; DMR, differentially methylated region; ENCODE, Encyclopedia of DNA Elements.

**Figure 2 nutrients-15-02167-f002:**
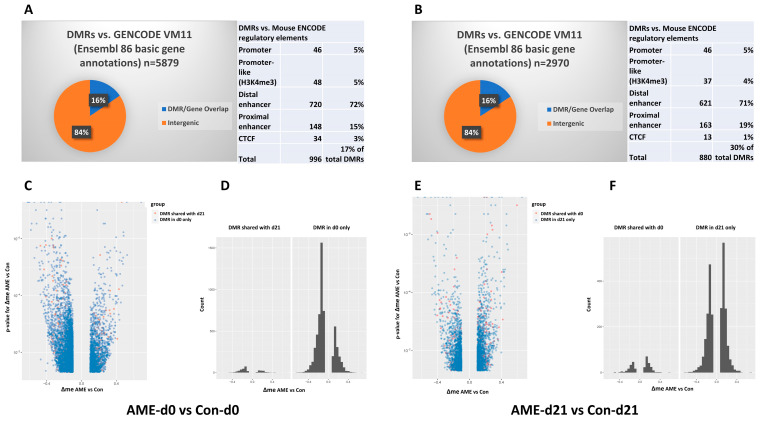
Effect of AME on DMRs in the livers of offspring (**A**,**C**,**D**) at d0 between AME-d0 and Con-d0 and (**B**,**E**,**F**) at d21 between AME-d21 and Con-d21. (**A**,**B**) The pie graphs show the percentage of DMRs overlapping genes in the intergenic regions by analyzing the DMRs against mouse gene annotations [GENCODE Version M11 (Ensembl 86)]. The tables show the number and percentage of DMRs overlapped with mouse ENCODE cis-regulatory elements, which are categorized into promoter, promoter-like (H3K4me3), distal enhancer, proximal enhancer, and CTCF binding sites. (**C**,**E**) Volcano plots depicting the p-value for Δme (the difference in average methylation between groups) for each DMR between AME and the control group. Hypo- and hyper-methylation refer to the loss and gain of DNA methylation in the AME group as compared to the age-matched control group. Each pink dot represents a DMR shared by both ages [shared between (AME-d0 vs. Con-d0) and (AME-d21 vs. Con-d21)] and each blue dot represents the unshared DMRs between ages. (**D**,**F**) Count the number of DMRs. AME, adverse maternal environment; Con, control; DMR, differentially methylated region; ENCODE, Encyclopedia of DNA Elements.

**Figure 3 nutrients-15-02167-f003:**
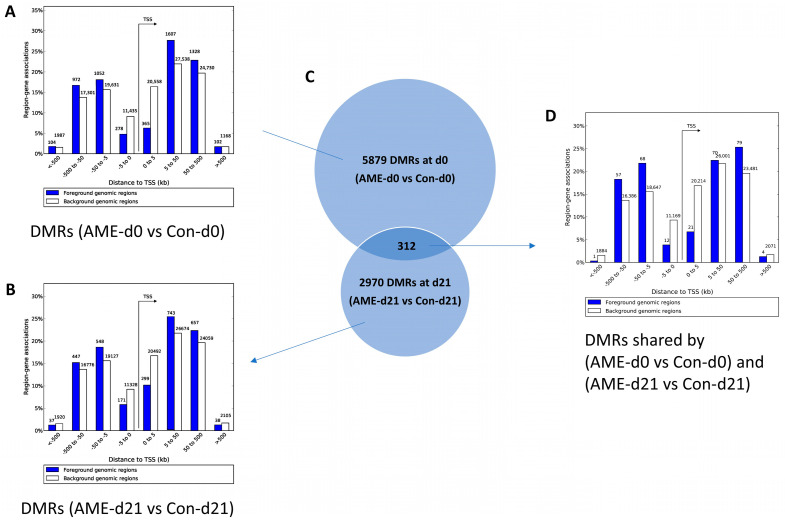
Effect of AME on the distribution of DMRs relative to the nearest TSSs. AME-resulted DMRs in the offspring livers at day 0 of life (**A**) and d21 (**B**) are binned by orientation and distance to the single nearest gene TSS within 1 Mb into 0–5 kb, 5–50 kb, 50–500 kb, and >500 kb at both upstream and downstream of the TSSs using the GREAT tool. (**C**) Venn diagram showing the number of AME-induced DMRs at day 0 of life (AME-d0 vs. Con-d0), at day 21 of life (AME-d21 vs. Con-d21), and those shared by both ages. (**D**) Distances to the nearest TSSs of AME-induced DMRs shared by d0 and d21. AME, adverse maternal environment; Con, control; DMR, differentially methylated region; TSS, transcription start site.

**Figure 4 nutrients-15-02167-f004:**
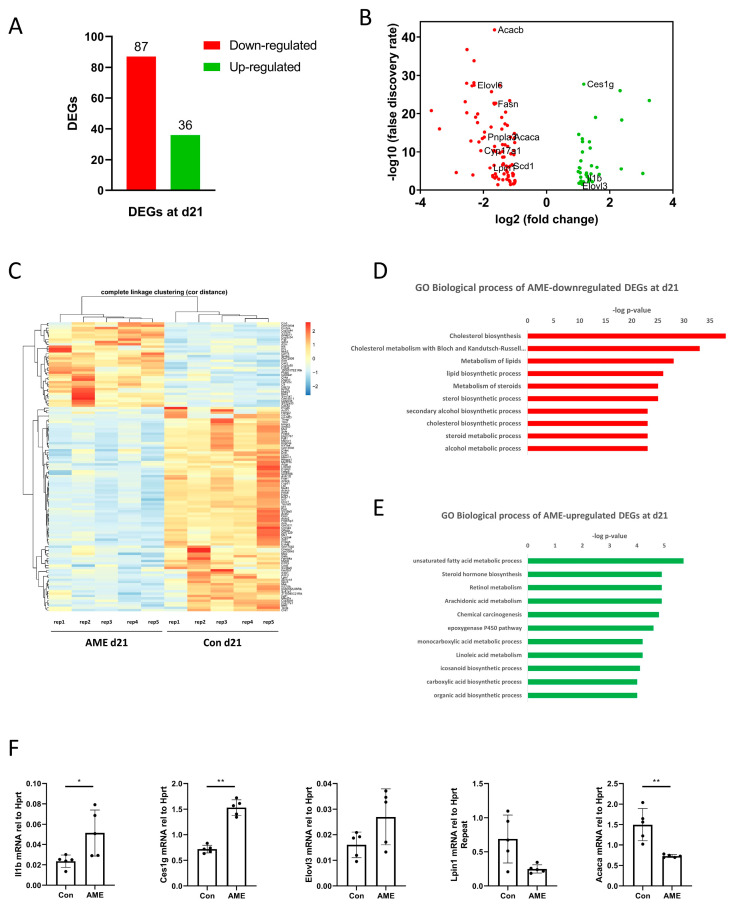
Effect of AME on the hepatic transcriptome measured by RNA-seq in offspring at d21**.** (**A**) The number of DEGs down-regulated (red) and up-regulated (green) by AME compared to the controls in offspring at d21 (n = 5). (**B**) Volcano plot depicting AME-induced hepatic DEGs in offspring at d21 compared to controls measured by RNA-seq [absolute log2 (fold change) > 1, false discovery rate < 0.05]. The downregulated DEGs are in red, and the upregulated DEGs are in green. The names of key genes are annotated. (**C**) Heatmap showing the expression levels of AME-induced DEGs with hierarchical clustering (Pearson correlation distance metric with complete linkage clustering), with each column representing one biological repeat (rep) (n = 5) in d21 offspring. (**D**) The top significantly enriched GO biological processes in AME-downregulated DEGs and (**E**) in AME-upregulated DEGs. (**F**) Real-time RT-PCR quantification of the genes involved in the significantly enriched GO biological processes. * *p*-value < 0.05; ** *p*-value < 0.01. AME, adverse maternal environment; Con, control; DEG, differentially expressed gene; GO, Gene Ontology; rep, repeat.

## Data Availability

Data that support the findings of this study have been deposited in GEO.

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
