# Peer review of "Adverse Maternal Environments Perturb Hepatic DNA Methylome and Transcriptome Prior to the Adult-Onset Non-Alcoholic Fatty Liver Disease in Mouse Offspring"

_nutrients, 2023, doi:10.3390/nu15092167_

Round 1
Reviewer 1 Report
In the present work, the authors aim to identify differentially methylated regions in DNA from a mouse model of NAFLD in order to identify those regions altered before the onset of NAFLD development.
The work is interesting and important as it identifies gene regions with potential to be used as a predictive tool for the development of NAFLD.
Although the work is complex due to the volume of information, it is well written and easy to understand.
Minor comments
- In the abstract, authors should delete the section headings
- The authors should improve the colours and resolution of the images 2 C and 2E, the colour of the pink dots is not visible.
- Could the authors represent the results described in lines 200 to 202 in Figure 1?
Reviewer 2 Report
In the manuscript by Fu et al, the authors look at the differences in the epigenomic landscape, particularly the DNA methylation and its consequent effect on transcriptome upon exposure to high fat/high sugar diet and AME and its contribution to the development of NALFD in the offsprings.
While the study is relatively well-executed, following are my suggestions to improve the scientific merit of the manuscript.
a) In the current format, the study design suffers from inadequacy to delineate the effects of diet from environmental stress. Perhaps the authors can look for changes in the transcriptome for strongest DGEs by qpCR in animals just fed with the western diet without any additional stress.
b) The DMRs reported in the study map to the cCREs many of which are the enhancers or enhancer-like elements. The authors can potentially use publicly available datasets for long-range DNA interactions in WT mouse, to comment on what these elements interact with and if that overlaps with (and explain) the transcriptomic changes observed.
c) It will be interesting to see what effect this paradigm had on female offsprings. Even though it is not within the scope of the manuscript to perform these experiments for the female offsprings ., it is recommended to include the BW and TG changes observed in female offsprings in the Supp Fig 1.
d) The authors should provide the Correlation coefficient/PCA plot to show how the replicates (for the RRBS and RNA seq) behave.
e) The data availability statement lack the information about the raw sequencing data access on GEO database which should be included in the final manuscript (and also for reviewers).
